# Deep and Transfer Learning Approaches for Pedestrian Identification and Classification in Autonomous Vehicles

Alex Mounsey, Asiya Khan *[ID] and Sanjay Sharma

School of Engineering, Computing and Mathematics (SECaM), University of Plymouth, Plymouth PL4 8AA, UK; alex.mounsey@postgrad.plymouth.ac.uk (A.M.); Sanjay.sharma@plymouth.ac.uk (S.S.)
* Correspondence: Asiya.khan@plymouth.ac.uk; Tel.: +44-1752-585123

**Abstract:** Pedestrian detection is at the core of autonomous road vehicle navigation systems as they allow a vehicle to understand where potential hazards lie in the surrounding area and enable it to act in such a way that avoids traffic-accidents, which may result in individuals being harmed. In this work, a review of the convolutional neural networks (CNN) to tackle pedestrian detection is presented. We further present models based on CNN and transfer learning. The CNN model with the VGG-16 architecture is further optimised using the transfer learning approach. This paper demonstrates that the use of image augmentation on training data can yield varying results. In addition, a pre-processing system that can be used to prepare 3D spatial data obtained via LiDAR sensors is proposed. This pre-processing system is able to identify candidate regions that can be put forward for classification, whether that be 3D classification or a combination of 2D and 3D classifications via sensor fusion. We proposed a number of models based on transfer learning and convolutional neural networks and achieved over 98% accuracy with the adaptive transfer learning model.

**Keywords:** pedestrian identification; classification; autonomous vehicles; CNN; transfer learning

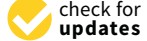



## 1. Introduction

Autonomous vehicles are becoming increasingly prevalent on roadways around the world; a study conducted in 2020 by Mordor Intelligence [1] reports that "the autonomous (driverless) car market was valued at USD 20.97 billion in 2020" and is projected to increase by 22.75%, to USD 61.93 billion by 2026. While consistent and significant technological advancements are being made in related fields, confidence in autonomous systems for use on roadways is declining. AAA reported in 2018 [2] that 73% of adults in the United States claim to be "too afraid" of allowing a vehicle to autonomously control itself—this is an increase of 10% from a similar study conducted one year prior.

Eliminating the human element of vehicular control, of course, subsequently eliminates the risk of traffic collisions resulting from human error. Furthermore, the occupants of autonomous vehicles are free to spend travel time recreationally or occupationally; this is especially beneficial considering the increasing congestion on roadways within major settlements, alongside a growing world population.

In the event that most, if not all, vehicles on roadways possess fully autonomous capabilities, it would be possible for a system to be implemented wherein these vehicles communicate with one another by sharing information on hazards ahead and manoeuvres they wish to perform. The resulting improvements to travel efficiency would likely have a cascading effect through iterative increases to speed limits.

Additionally, the Mobility-as-a-Service (MAAS) market is likely to see an increase in potential as autonomous vehicles acquire mass-adoption. Fully autonomous MAAS would hypothetically enable individuals to, rather than owning a personal vehicle, lease a vehicle for each journey they embark upon, similar to how companies such as Uber and Lyft currently operate, however in this case, without the need for a driver. Alternatively,

those who do opt to purchase their own autonomous vehicle would have the opportunity to lease the vehicle out when not in use, providing an additional stream of income.

In an autonomous driving system, the safety of vehicle occupants, as well as individuals in the surrounding environment, should be guaranteed. One recent study conducted by Najada and Mahgoub [3] revealed that approximately 80% of casualties resulting from vehicle-related accidents were pedestrians.

The safety of vehicle occupants and pedestrians can be achieved through collision avoidance warning systems (CAWS). A key component of CAWS is vehicular situational awareness, which can be facilitated through the use of different types of sensors. These sensors gather data pertaining to the vehicle's surroundings, which can then be processed, with useful information being extracted. LiDAR, RADAR, and camera sensors are the three most prominent sensors currently in use.

In this paper, the use of cameras and computer vision in the scope of pedestrian detection and classification, touching on methods by which LiDAR can be used to improve such a system through sensor fusion are investigated. Here, pedestrian detection can be defined as the process of determining whether an image, generally a frame extracted from a video sequence, contains pedestrian instances. A successful system should be able to leverage computer vision technologies in order to extract the specific locations of any pedestrians in the frame [4], the results of which are usually in the form of bounding boxes encapsulating individual pedestrian instances.

Machine learning models are generally bespoke, designed with a specific use-case in mind. While the performance of these models can be exceptional, the training process requires a substantial amount of labelled data, which can be incredibly time-consuming. ImageNet [5] is an example of such a dataset, consisting of over 14 million images across thousands of classes. Models trained using ImageNet may be exceptional at differentiating between a wide variety of classes, however, applying such a model to a more specific use-case would likely result in a significant loss of performance. Hence, the motivation to make use of transfer learning in this paper to reduce computer complexity and enable the transfer of learning from a previously trained model.

Identifying and localizing pedestrian shapes in images has, perhaps, been one of the greater challenges facing computer vision researchers over the past decades [6], largely due to the variable appearance of the human body and variations in illumination, occlusion, and poses [7]. Recently, however, with the advent of increasingly powerful and compact hardware, pedestrian detection systems have taken great strides in terms of efficiency and accuracy [8–10].

There are two primary methods of achieving pedestrian classification through computer vision: deep learning [11] and machine learning [12] based methods; both approaches follow similar computational pipelines. First, candidate regions must be identified—this can be achieved through the application of either a sliding window, or some more complex region proposal algorithm [13,14]. Once candidate regions are identified, feature extraction is applied to these regions to obtain an accurate classification on the basis of subsequent classification algorithms.

In 1999, Lowe proposed a visual recognition system [15] which makes use of local features which are scale-invariant and partially invariant to changes in illumination. This publication is indicative of researchers' shift in focus at the time, from attempts to reconstruct objects as three-dimensional objects [16], to feature-based object recognition. Soon after, Viola and Jones published a real-time facial recognition framework [17] in the form of a binary classifier consisting of numerous, weaker, classifiers which are trained using Adaboost [18]. Viola and Jones later went on to propose a pedestrian detection algorithm which used motion and appearance information in order to detect a moving person [19]. Dalal and Triggs expanded this work [20] and proposed the use of Histogram of Oriented Gradients (HOG) as a feature extractor, with the resulting HOG features being fed into a linear Support-Vector Machine (SVM) [21] classifier. This HOG-SVM combination is capable of differentiating between regions which contain pedestrians and those which

do not. The resulting reduction in the number of false positives was over an order of magnitude, compared to the best performing Haar wavelet detector at the time [22]. While HOG-SVM offers exceptional performance in classification tasks, it fails to achieve a low mean average precision [23].

In 2008, Felzenswalb et al. utilised the HOG-based detector in their multiscale Deformable Part Model (DPM) [24] which deconstructs objects into groups based on pictorial models [25]. The DPM was suggested to be state-of-the-art at the time, outperforming other methods of object detection, such as template matching.

McCulloch and Pitts first proposed the McCulloch-Pitts (MCP) model in 1943 [26], which is widely accepted as the genesis of Artificial Neural Networks. In 1980, Fukushima introduced Neocognitron, a hierarchical, multilayer Artificial Neural Network which was designed for use in handwritten character recognition and similar pattern recognition tasks. The model consisted of several pooling and convolutional layers, which provided the ability to learn how to identify visual patterns in images. LeCun et al. inspired from Neocognitron proposed the concept of Convolutional Neural Networks (CNNs) which utilize error gradient, yielding impressive results in a range of pattern recognition applications [27–29].

CNNs [30] are perhaps the most prevalent application of deep learning for computer vision tasks, as they have proven to be exceptionally well-suited for tackling object detection problems, in part due to their ability to extract discriminative features. CNNs are composed of three different types of neural layers: convolutional layers, pooling layers, and fully connected layers. In the context of computer vision tasks, Yosinski et al. [31] deduced that the lower layers (i.e., convolutional and pooling) act in a similar manner to conventional computer vision-based feature extractors such as edge detectors, while the final, fully connected layers, are more task-specific. In [32] authors showed that CNNs outperformed both HOG descriptor and Haar-classifier.

As discussed in earlier sections, deep learning and machine learning models require significant volumes of data for use during training. This was identified in 2001 in a research report published by Gartner [33], which alluded to the impending surge of big data. An onboard pedestrian detection system is proposed in [34] based on 2D and 3D cues. Just under a decade later, the ImageNet database was introduced by Deng et al. in 2009 [5]. Authors in [35] propose a dataset that includes challenges related to dense urban traffic, based on their dataset they propose a fusion framework for multi-object detection. The advent of larger datasets such as ImageNet required more capable deep learning models and, in 2012, Krizhevsky et al. introduced AlexNet [36]: a breakthrough in CNN architecture which makes use of the Rectified Linear Units (ReLU) activation function which provided a sixfold reduction in training time, compared to the TanH activation function which, at the time, was standard. Additionally, AlexNet has the capability of being trained across multiple GPUs simultaneously, which enabled more complex models to be produced and was a key enabler of the significant reduction in training time.

Transfer learning (TF) aims to provide a middle ground, where knowledge acquired from larger datasets can be used in conjunction with smaller, domain-specific, datasets in order to improve performance in subsequent domain-specific tasks. In this context, prior knowledge can be model weights or low-level image features which describe what is being classified such as edges, shapes, corners, pixel intensity, etc.

Therefore, the models produced in this work make use of transfer learning as it enhances the performance of the proposed CNN model with the VGG-16 architecture, proposed by Simonyan and Zisserman in 2014 [37]. The VGG-16 CNN model improves upon the work carried out for AlexNet by switching the $11 \times 11$ and $5 \times 5$ kernel-sized filters with two consecutive $3 \times 3$ filters in the first two convolutional layers.

The main contributions of this paper are threefold—(i) a review of CNNs in pedestrian classification, (ii) classification models trained on CNNs and transfer learning and (iii) a pre-processing system with LiDAR point cloud with applications in a 3D object classification model.

The rest of the paper is organized as follows. Section 2 presents the review of CNNs. The models developed are explained in Section 3. Results and discussions are elaborated in Section 4 with a conclusion in Section 5.

## 2. Review of CNNs for Pedestrian Recognition

R. Hecht-Nielsen [38] described neural networks as "a computing system made up of a number of simple, highly interconnected processing elements, which process information by their dynamic state response to external inputs". The review presented here expands on the CNN and deep learning principle presented in [39,40] in the context of AlexNet [36]. They are modelled to mimic the human brain in order to recognize patterns. This is achieved through numerical input vectors that describe real-world information such as images and text, from which an output response can be generated. In the context of pedestrian classification, once a candidate region has been recognized through recognition techniques, it can be classified through the use of a neural network which allows for an appropriate response to be made by the vehicle.

Convolutional Neural Networks (CNNs) mostly used in the computer vision field. CNNs are structured in a three-dimensional layers and processes information first through "convolution", layer where small portions of data are analysed in order to create a "feature map", before passing it to "pooling" layer. Here, each feature of the data set has its dimensionality reduced while retaining the most relevant information. This next section covers the related theory behind CNNs.

### 2.1. Single Layer Perceptrons

Perceptrons, sometimes referred to as "linear binary classifiers", are a form of supervised classification algorithm that can be used to determine the classification of a given input. If neural networks are considered to be computational representations of the human brain the perceptrons act as individual neurons, which take the form of a single-layer neural network and consist of four key elements: input values, weight and bias, the net sum, and an activation function.

Input values are multidimensional vector values that are fed into the perceptron in order to be processed. The input values are multiplied by a weighting parameter, which is indicative of an individual input's influence over the output value. The sum of the weighted input values is referred to as the "net sum" or "weighted sum", and can be calculated with the following equation:

$$s = \sum_{i}^{m} w_i I_i,$$ (1)

where $s$ is the weighted sum, $m$ is the number of inputs, $w$ represents the weight for each input, and $I$ represents the value of each input. Once a weighted sum has been calculated, it is then applied to the activation function, which normalizes it. In simple perceptron models, the activation is a step function. See Figure 1.

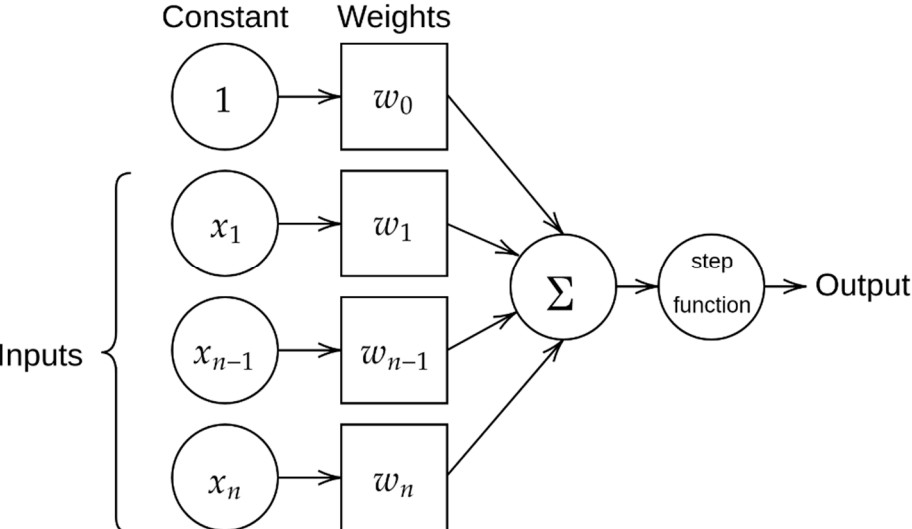

**Figure 1.** A simple perceptron model.

### 2.2. Multi-Layer Perceptrons

Multi-layer perceptron (MLP) is simply another way of referring to a neural network and consists of a collection of individual single-layer perceptrons (SLP) arranged into distinct "layers". The most basic form of MLP consists of three layers: an input layer, output layer, hidden layer. The input and output layers serve the same purpose as in an SLP, the hidden layer is where most of the MLP's computation is performed.

MLPs allow for non-linear classification, such as XOR functions, which is not possible with SLPs as they are not capable of modelling feature hierarchy. It is for that reason that SLPs generally only find use as building blocks for MLPs, which have been shown to approximate non-linear functions. Furthermore, SLPs simply use the step function as an activation function, whereas MLPs can use more complex activation functions which enable the classification of items into multiple labels as well as to provide probability-based prediction.

### 2.3. Activation Functions

Activation functions are mathematical equations that are not only used to determine the output of the individual perceptrons, but also the accuracy, computational efficiency during training, and the output of a deep learning model in its entirety. Additionally, selecting an appropriate activation function for the task the neural network is attempting to perform is crucial, as they have a significant influence over the network's ability to converge and the speed at which it can converge. Examples of activation functions include the binary step, linear, sigmoid, TanH, rectified linear unit (ReLU), SoftMax and back propagation. ReLU is the most frequently used activation function due to their simplicity where positive values are treated linearly, and negative values are assigned a value of zero [41].

### 2.4. Hyperparameters
#### 2.4.1. Hidden Layers and Units

Hidden layers are layers within a neural network that lie between the input and output layers. Increasing the number of hidden layers has the potential of increasing the accuracy of a model, however as more hidden layers are added, computational requirements will increase yet will yield diminishing returns on the error function.

Not having an adequate number of hidden layers on the other hand will result in poor generalization and unreliable predictions, so it is important to strike a balance in the selection of number of hidden layers.

### 2.4.2. Dropout

Dropout is a technique used during the training of a model in which certain nodes are deactivated so that it does not become overwhelmed with information, which can isolate nodes that may not be contributing to an improved error function, which in turn should produce a more efficient model.

### 2.4.3. Activation Function

Activation functions, which have been discussed earlier in this section, can be added to any point of a neural network and there is no limit to the amount that can be added which again results in the process of determining a suitable balance between the number of activation functions and the overall efficiency of the model.

### 2.4.4. Learning Rate

The learning rate determines the strength of changes made to weights during the process of backpropagation. A lower learning rate results in smoother convergence at the cost of an increase in training time and a higher learning rate will have opposing effects, which means the appropriate learning rate will be model-specific.

### 2.4.5. Epochs and Batch Size

The number of epochs represents the number of instances that the training dataset is fed into a neural network during the training process. Increasing the number of epochs will increase the accuracy to a certain extent, after which overfitting will start to occur, and training accuracy can decrease. Batch size controls the percentage of the dataset to be exposed to the network through each iteration (epoch) of the training process, which can reduce the over generalization of the model.

### 2.4.6. Optimisation Algorithm

Optimisation algorithms are those that attempt to minimize the error function of a model. There are two subcategories of optimization algorithms: "first-order" (e.g., gradient descent), which adjust the loss function with respect to given parameters, and "second-order", which use what is known as the 'second order derivative' or "Hessian" to adjust the loss function.

First order optimizations are easier to compute and require less computational time to converge reasonably well with larger data sets. Second order derivatives are only able to outperform first order optimizations when a second order derivative is known, otherwise they are more computationally intensive and take longer to execute.

### 2.4.7. Momentum

Momentum is the process of tracking changes made to a model and the direction of those changes, which can be used to influence subsequent changes so that they follow the same direction, towards a lower error function.

## 3. Proposed Pedestrian Classification Models

Pedestrian detection can be described as a binary classification problem, where the goal is to predict whether candidate regions contain an instance of a pedestrian (positive sample, denoted as 1) or not (negative sample, denoted as 0).

All models and software produced for this project were developed using Python 3.8. TensorFlow 2.3.0, Keras 2.4.3, and scikit-learn 0.24.2 were used in the development of deep learning models. Scikit-learn also found use in the development of our 3D pre-processing system, in the creation of the DBSCAN clustering algorithm and RANSAC regressor. TensorBoard was used during the training of CNN models in order to monitor training progress in real-time.

The overall concept of our proposed method is presented in Figure 2.

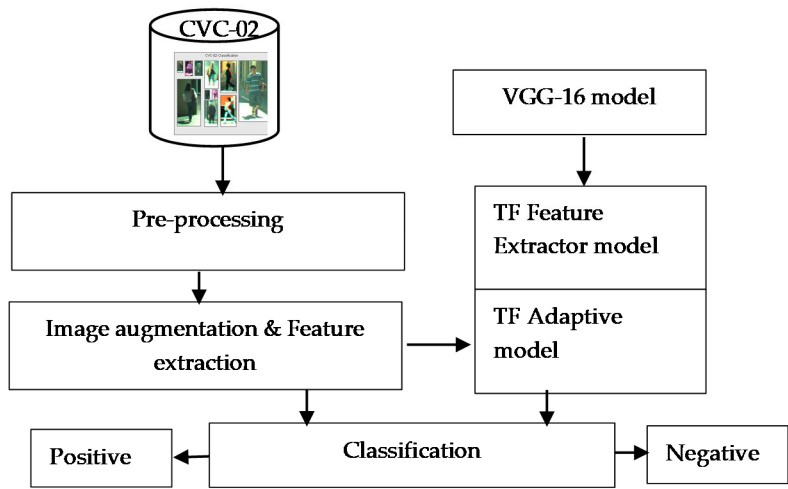

**Figure 2.** Concept of our proposed method.

In Figure 2, at the pre-processing stage the training and validation images have their pixel values scaled from (0 to 255) to (0 to 1) upon import. In the next block, image generators augment the imported images before supplying them to our models. This includes minimal random rescaling, as well as random zooming and shear-transformations by a factor of 0.3. In the TF adaptive model block, we do not use bottleneck features as we use image generators, whereas the TF feature extractor model makes use of the VGG bottleneck features.

This section presents the datasets used, data preparation and image augmentation parameters.

### 3.1. Datasets

#### 3.1.1. CVC-02 Dataset

The CVC-02 [34] dataset was used during the development of the pedestrian classification systems for an autonomous vehicle. Images provided by the CVC-02 dataset have been recorded in urban scenarios in and around Barcelona, Spain. Images have been recorded using a colour stereo camera with a resolution of 640 × 480 pixels and 6 mm focal length. Specifically, the classification subset which consists of 3140 positive and 6175 negative samples (in which pedestrians are present, and are not present, respectively) were prepared for this paper; these images have also been rescaled to a size of 64 × 128 pixels and have been split into training, validation, and testing dataset. Table 1 presents the overview of the CVC-02 data splits used in the development of pedestrian classification models.

**Table 1.** Overview of CVC-02 data splits.

| Data Split | Total Samples | No. of Positive Samples | No. of Negative Samples |
|------------|---------------|-------------------------|-------------------------|
| Training   | 3000          | 1500                    | 1500                    |
| Validation | 1000          | 500                     | 500                     |
| Testing    | 1000          | 500                     | 500                     |

#### 3.1.2. NuScenes Dataset

The NuScenes [42] dataset is a large-scale dataset provided for use in autonomous driving research and has been used here in the development of a 3D LiDAR pre-processing system. The entire database consists of 1000 20-s scenes recorded in Boston, United States and Singapore under challenging driving conditions. Each scene contains data captured using 6 cameras, 1 LiDAR sensor, 5 RADAR sensors, a GPS, and an IMU. The 10 scenes which is a subset of the main dataset are used here to show the effectiveness of this approach.

### 3.2. Image Augmentation

When training deep learning models such as CNNs, the quantity of available data may be a limiting factor for model performance—this can be somewhat alleviated through the application of image augmentation, which artificially creates a new data samples based on original samples. While there are appropriately sized datasets available for the training of pedestrian classification models, image augmentation can also be used to apply artificial transformations to input images such as rotation, scale, shearing, zoom, etc. These augmentations are employed in an effort to mimic noise, variations in illumination, and variations in fundamental image properties (e.g., scale, rotation).

It is preferable that the types of augmentation, and the intensity to which they are applied, strike a balance between providing a suitable amount of noise and variation, which can increase the complexity of the data, and preserving the image features, which is crucial in a model being able to generate accurate predictions. Figure 3a,b demonstrates examples of suitable and unsuitable image augmentation. The images can be improved by pre-processing techniques such as wavelet-based de-noising to remove the noise. Note that in the unsuitable data, parts of the pedestrian have been cropped out of the image—this, of course, will not provide the model with representative information. Furthermore, the augmentations should make sense in the context of the problem being solved; applying a vertical flip to a pedestrian dataset is ill-suited as it is unlikely that a deployed model will encounter a pedestrian in such a position, although edge cases will exist.

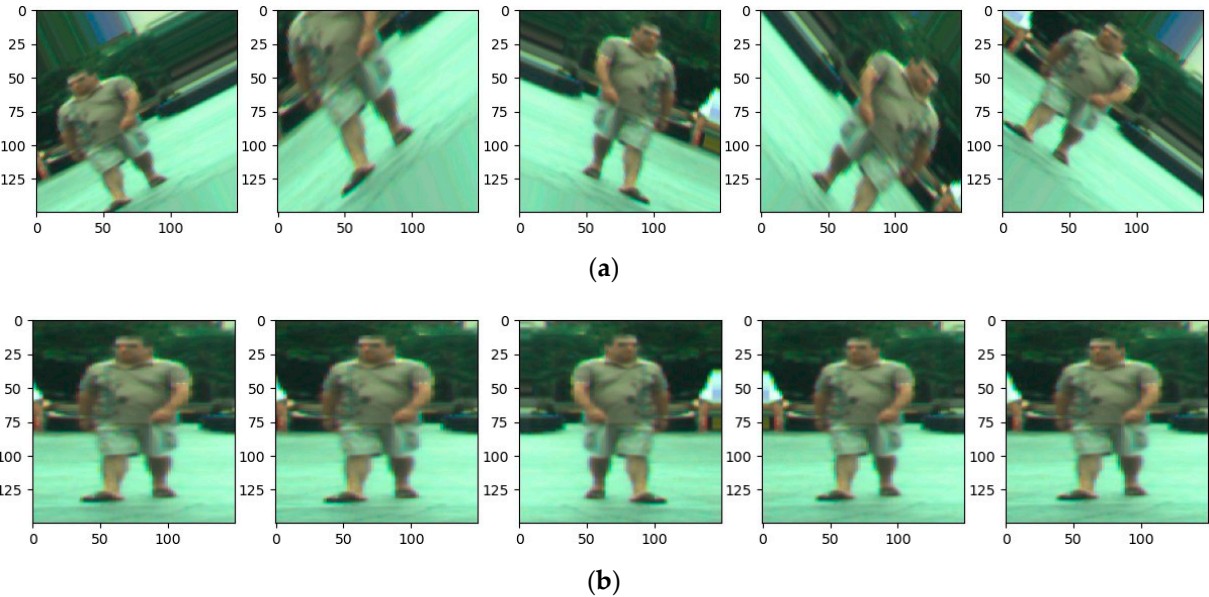

**Figure 3.** (**a**) Unsuitable image augmentation. (**b**) Suitable image augmentation.

All models trained during this project made use of identical image preparation techniques. Augmentation was applied to all models, except for the model described in Section 3.5.1, which takes the VGG-16 bottleneck features as input.

Image dimensions are first changed to 150 × 150, with three channels corresponding to the RGB colour space. Image pixel values are then normalized, from values between 0 and 255, to values between 0 and 1. The normalization of pixel values is to increase the rate at which models learn; large pixel values can disrupt or slow down the process. The original sample labels, which are strings ("positive" or "negative") are encoded to numerical values (1 or 0, respectively), which is more suitable for machine learning techniques.

Once the data has been prepared, image augmentation is applied. Keras' Image-Data Generator() method was used to facilitate this augmentation. The parameters of this augmentation are as follows: a shear range of 0.1, a zoom range of 0.1 with the fill mode set

to "nearest" (new pixels are set to the nearest neighbouring pixel values), with horizontal flipping enabled. Figure 3b illustrates the effects of these parameters on training data.

During the development of the models, it was found that prior configurations for image augmentation resulted in models that were unable to identify pedestrians in the majority of samples (REF RESULTS). After investigation, it was found that the augmentation being applied was too intense, resulting in images which contained pedestrians who could not be identified by the model. The offending parameters included: A zoom range of 0.3, rotational range of 90 degrees, width and height shift ranges of 0.2, and shear range of 0.2. This faulty image augmentation is illustrated in Figure 3a.

### 3.3. Rudimentary CNN Classifier

The first CNN model produced in this paper was developed to serve as a benchmark for comparison against transfer learning-based models. The architecture of this model is relatively simple, consisting of three blocks of convolutional layers, the output of which are flattened into feature maps of shape $17 \times 17 \times 128$, which are then processed by two dense layers as shown in Figure 4.

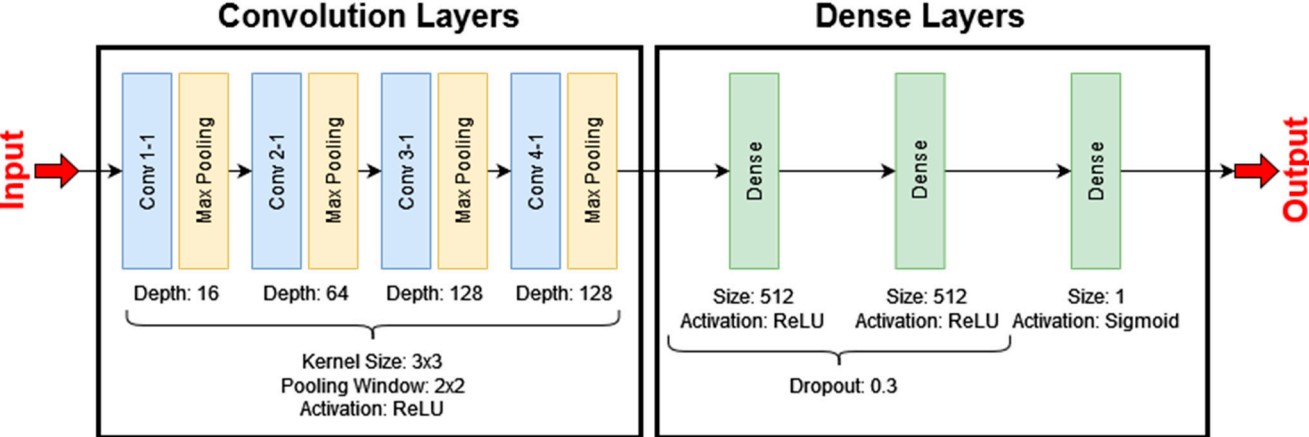

**Figure 4.** Rudimentary CNN model architecture.

In the convolutional layers, a kernel size of $3 \times 3$ has been used—this is based on the kernel filter size used in the VGG-16 architecture which, in-part, resulted in an improvement of performance over the AlexNet architecture (see Section 3.4). The final output of the dense layers is a confidence score for each class, indicating how confident the model is that a sample does or does not contain a pedestrian instance.

Training of the first rudimentary model spanned 100 epochs, with a batch size of 30. 100 iterations per epoch was used to cover the training data consisting of 3000 samples (Table 1), with each sample being used to generate 30 additional samples via image augmentation. As the augmentation of validation data results in only 20 images per original sample (scaling is the only augmentation applied to the validation set), the number of validation steps per epoch has been set to 50, such that the model is able to validate on all available samples.

As observed in Figure 5, during training, the model began to overfit after the fourth epoch, as illustrated by the increase in loss on validation data. This means that the model was not capable of generalizing unseen data. However, decent results with a final validation accuracy of 96% were obtained.

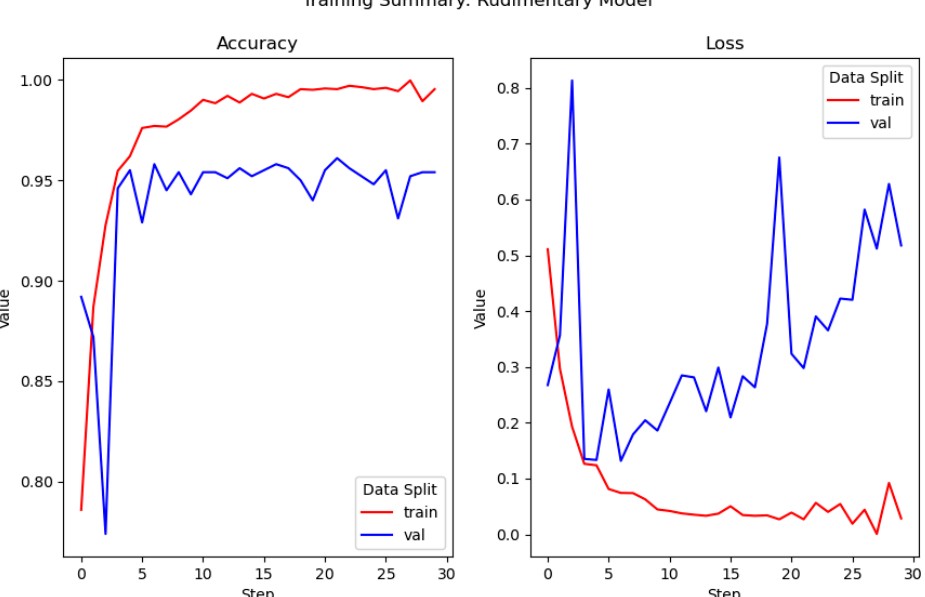

**Figure 5.** Training summary of the rudimentary CNN model.

Figure 5 is composed of two training summary plots. The left plot details the accuracy of the model during training, i.e., what percentage of data samples can it accurately identify; the red line corresponds to the model accuracy on training data, while the blue line corresponds to the model accuracy on validation data. The right plot details the model loss during training. Loss is a measure of how far an estimated value is from its true value. In all models described in this paper, binary cross entropy is used as a loss function.

### 3.4. Rudimentary CNN Classifier with Regularization

In an attempt to improve upon the performance of the previous model, a second model has been developed which incorporates regularization in the fully connected layers. This is achieved via the addition of two dropout layers, one after each of the first two dense layers. A fourth convolutional block, consisting of a convolution and max-pooling layer is also added. The parameters of this new block (depth, filter size, and activation function) are identical to those in the preceding layer (Figure 6). The reasoning behind the addition of a fourth convolutional block is to enable the model to extract more features from input samples, resulting in an improvement of performance.

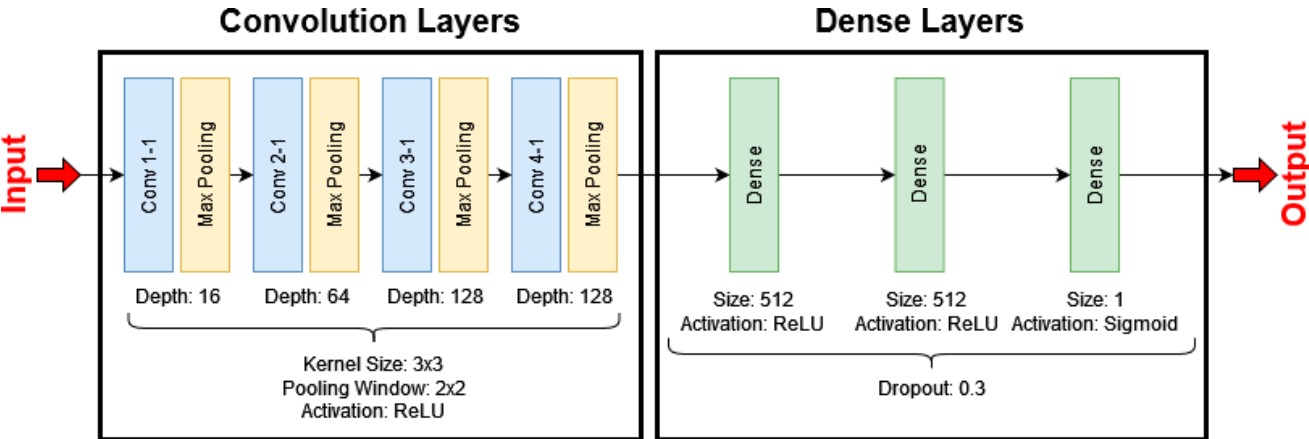

**Figure 6.** Rudimentary CNN model architecture with regularization.

Dropout layers enable regularization in deep learning models, which simulates the use of multiple architectures during training. Dropout layers randomly mask a fraction of units' output by setting their values to zero. Dropout is a computationally inexpensive and effective technique of reducing the rate of overfitting in a model and improves its generalization error.

Here, a second rudimentary model which makes use of regularization through the addition of dropout layers is presented. In this case, a dropout of 0.3 is applied to the output of the first two dense layers. The result of this is that 30% of the units in these dense layers are masked (i.e., 30% of the units in each layer have their output nullified; set to zero). Additionally, a fourth convolutional block has been added. This enables the model to extract more features from input samples, with the goal of an increase in performance over the original model.

The regularized model is trained with the same hyper parameters as in the original model, spanning 100 iterations across 30 epochs. The addition of regularization and a fourth convolutional block, while resulting in similar validation accuracy (96%), reduces the validation loss considerably (approx. 0.51%, down to approx. 0.40%). Furthermore, the model begins overfitting later in the training cycle, and the rate of overfitting is less intense than that in the previous model. The training data indicates that this model is more suitable for use on unseen data than the former model, suggesting that this model is more capable of generalizing better (Figure 7).

**Figure 7.** Training summary of the rudimentary CNN model with regularization via dropout layers.

### 3.5. Transfer Learning Classification Models

VGG-16 [36] is a CNN architecture which specializes in computer vision recognition tasks and has been trained on the ImageNet dataset [5]. It takes $224 \times 224$ RGB images as input. The "16" in its name refers to the 16 composite layers, which are split into five convolutional blocks and a single fully connected block, that make up the architecture (Figure 8).

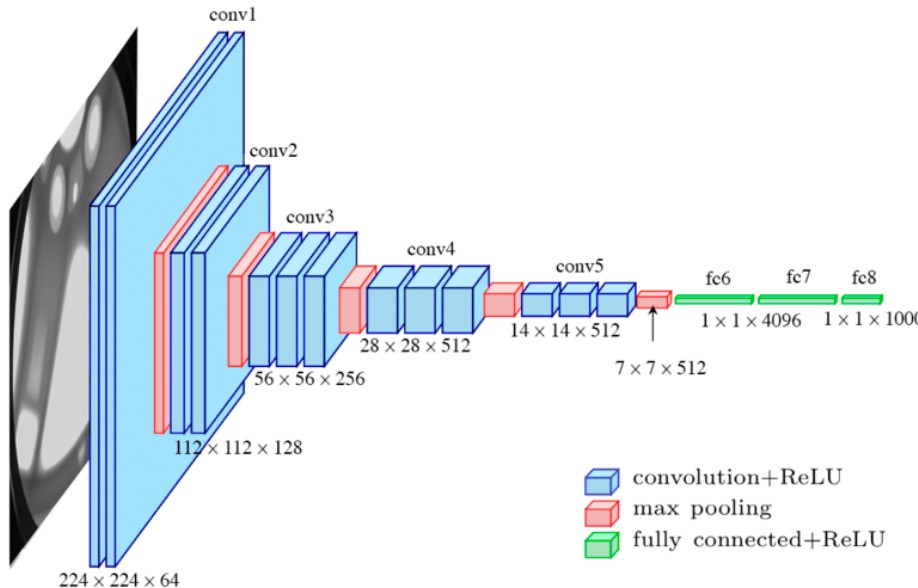

**Figure 8.** VGG architecture diagram [41].

The convolutional layers all make use of $3 \times 3$ convolution filters, with each convolutional block paired with a max pooling layer. The max pooling layers are applied over a $2 \times 2$ window with a stride of 2, for down sampling purposes. Note that, in blocks 3 and 4, VGG-16's use of $3 \times 3$ filters resulted in an improvement of performance over AlexNet and ZF-Net, which used $11 \times 11$ and $7 \times 7$ filters, respectively. It is also worth highlighting that $3 \times 3$ filters are the smallest dimensions that allow a model to learn directional features (e.g., up, left, centre, etc.).

Convolutional block 5 produces a feature map of shape $7 \times 7 \times 512$, which is flattened into a feature vector containing bottleneck features. These bottleneck features are then processed by a block of fully connected layers consisting of two 4096-channel layers and a single 1000-channel layer (with one channel for each class in the ImageNet dataset). Finally, a SoftMax activation layer normalizes the classification vector, ensuring that the sum of all probabilities is equal to zero.

In this sub-section, we describe three models developed via transfer learning: the first two of which "freezes" the convolutional blocks in order to preserve the original VGG-16 model weights, which are not updated during training. The original dense layers are replaced with bespoke dense layers which are more suited to the specific task of pedestrian classification. The second and third models make use of previously discussed image augmentation methods, while the first does not. Instead, aforementioned bottleneck features are provided to the model during training. These features must also be processed for subsequent classifications on new data.

The third model enables the updating of weights in convolutional blocks 4 and 5. These weights are initialized as those in the vanilla VGG-16 model. The dense layers are also replaced with those used in the previously described model.

In all three approaches outlined above, the dense layers are replaced. This is in-line with the findings of Yosinski et al. [31], described in the literature review: convolutional layers act as conventional computer vision feature extractors, which can be applied to a wide scope of tasks. The outputs of the final convolutional block take the form of 'bottleneck features' (Figure 9) which are the final activation maps prior, processed by the fully connected dense layers. The blurriness in Figure 9 can be improved by pre-processing the image.

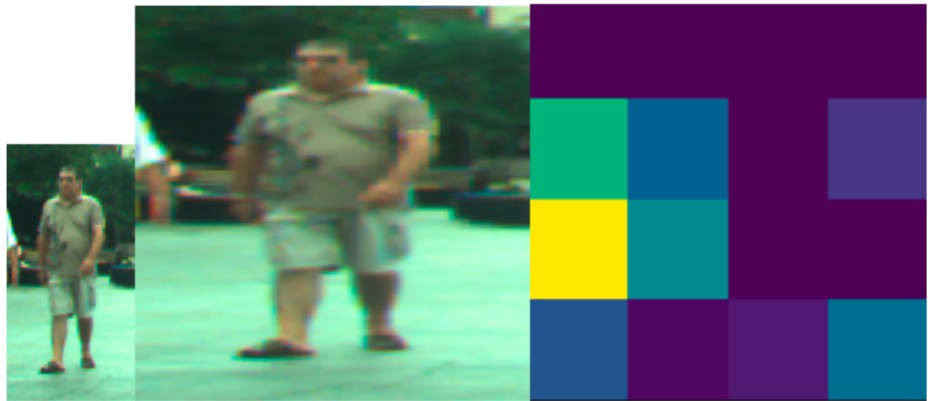

**Figure 9.** Visual representation of bottleneck features from the VGG-16 model.

### 3.5.1. VGG-16 Feature Extractor Model

This section describes the first of three models with the VGG-16 architecture used in the transfer learning model. In this model, all convolutional layers of the original architecture have been frozen; their weights will remain the same throughout the training process, not being updated as in the traditional process of training CNN models. The dense layers have been replaced: the new dense layers follow the same form as in the rudimentary models proposed in Section 3.3. Furthermore, no image augmentation has been applied to the data samples provided during training– this is to compare the results of image augmentation and will be compared to a subsequent model which does incorporate image augmentation. Instead, the vanilla VGG-16 model is used to generate bottleneck features for all training and validation data samples, these bottleneck features are then flattened for use as inputs for our model.

The hyper parameters used in the training of this model are identical to those used in the training of the rudimentary models described previously: 30 epochs of 100 iterations in training, with 50 iterations during validation. The model performance appears to have significantly improved over the aforementioned rudimentary models (Figure 10): validation loss, while still indicative of overfitting at the fifth epoch, has been reduced by approximately 30%. Furthermore, the training and validation accuracies are more closely correlated (a variation of approximately 2%) which suggests that the transfer learning model is more capable of generalizing on unseen data, than the rudimentary models.

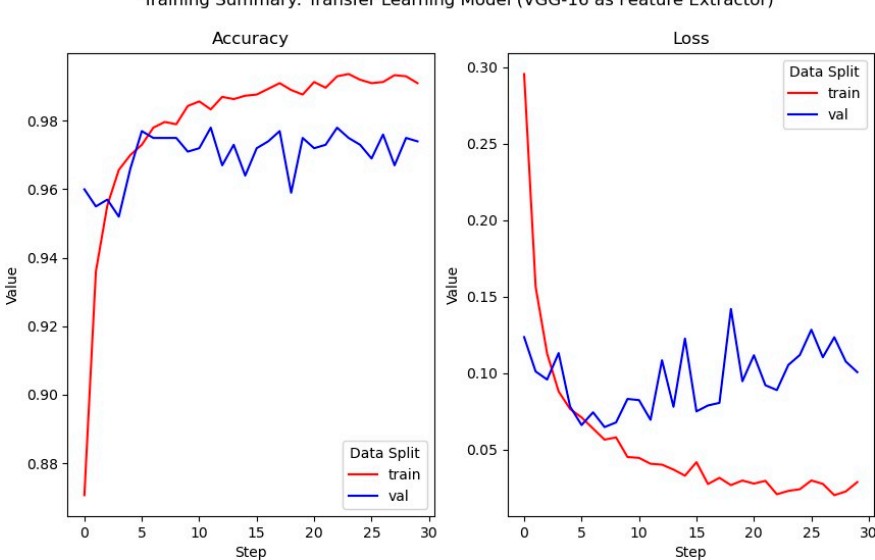

**Figure 10.** Training summary of the VGG-16 feature extraction model.

### 3.5.2. Applying Image Augmentation

In this model, image augmentation is applied to input samples—the process of which is identical to that in the rudimentary models (Section 3.3). The model has been trained for 100 epochs of 100 iterations, with 50 iterations for validation steps. Additionally, the learning rate is reduced (from $1 \times 10^{-4}$ to $2 \times 10^{-5}$) in order to avoid rapid and abrupt changes to weight values during backpropagation which may adversely affect model performance.

The training of this model spanned 100 epochs of 100 iterations, with a batch size of 32: the default batch size for Keras. The training summary of this model (Figure 11) is similar to that of the previous model, suggesting that the application of image augmentation did not provide significant improvements to model performance in this case.

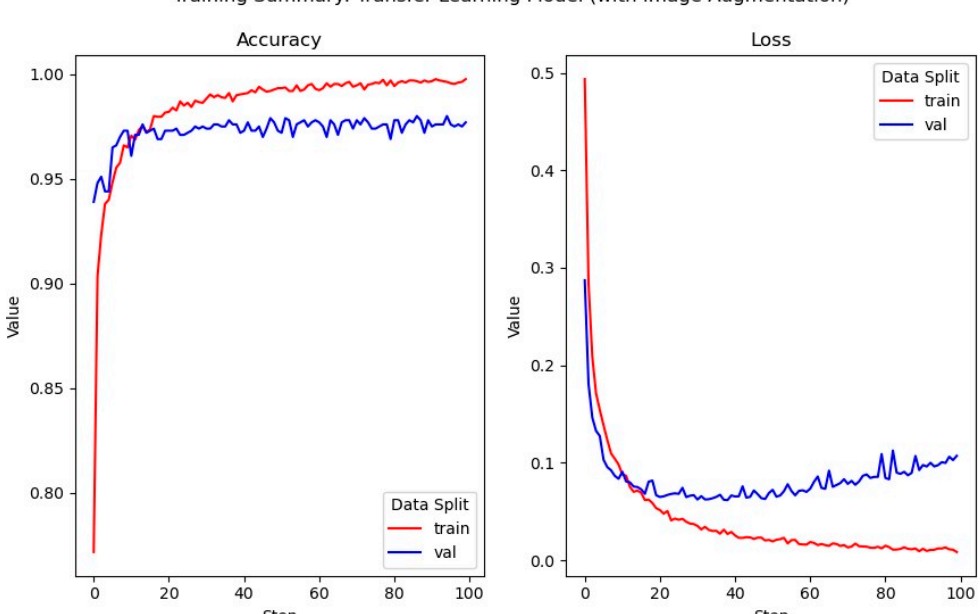

**Figure 11.** Training summary of the VGG-16 transfer learning model with image augmentation.

### 3.5.3. Adaptive VGG-16 Model

This section describes the third and final transfer learning model developed during this project. This model enables the updating of weights in convolutional blocks 4 and 5; blocks 1–3 remain frozen, preventing the weights of these blocks from being updated during backpropagation. The learning rate is further reduced from $1 \times 10^{-4}$ to $1 \times 10^{-5}$, again to prevent adverse effects on model performance as a result of intense adjustments to weight values and avoid the model from getting stuck in local minima. This model has been trained with the same image augmentation as in previously discussed models.

The training of this model, again, spanned 100 epochs of 100 iterations, with a batch size of 32. The training summary of this model (Figure 12) illustrates that, while validation loss appears to have increased from previous models, the validation accuracy has improved (approx. 98%). This is likely due to the model's ability to better understand the provided data though the adjustment of weight values during backpropagation.

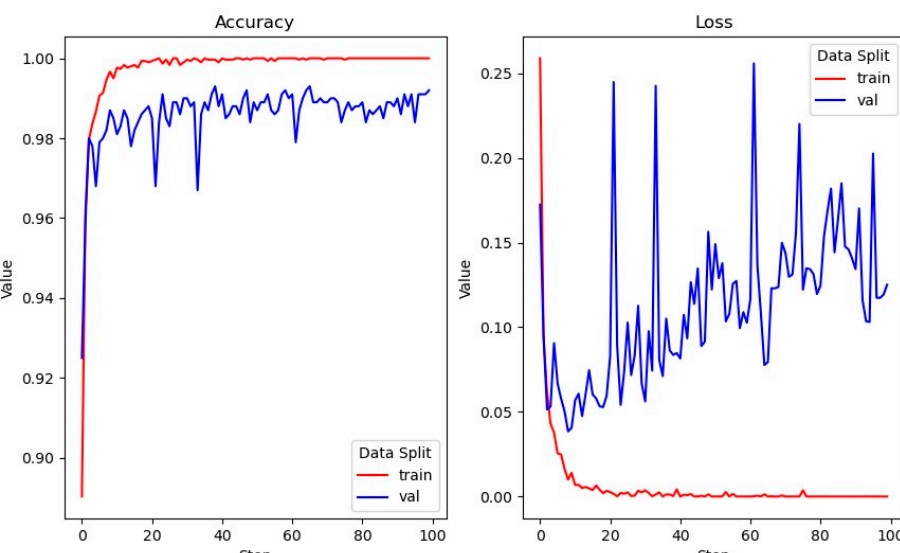

**Figure 12.** Training summary of the VGG-16 transfer learning model with convolution blocks unfrozen.

### 3.6. 3D Point Cloud Pre-Processing

In the development of the 3D pre-processing system described in this section, a number of tools and software were used in this paper to produce results. Most notably: Python 3.8, NumPy, Pandas, and the NuScenes devkit. Furthermore, Scikit-Learn facilitated the implementation of our RANSAC regressor and DBSCAN clustering algorithm.

#### 3.6.1. Identification and Removal of Ground Points

LiDAR point cloud data is inherently noisy and can contain large amounts of data which may not be pertinent to the task at hand. Examples of such points are those which correspond to the ground on which a vehicle is travelling. While information about the ground can be useful in the identification of road markings, it does not provide meaningful information in the context of pedestrian detection (there are specific use-cases, an example of which being the identification of regions of high pedestrian activity such as crossings, however). Removing ground points not only removes points which may skew the results of further operations, it also significantly improves the computational time and effort required to process the entire dataset.

#### 3.6.2. Data Preparation

A crucial and effective method of removing irrelevant points from the provided point clouds is to identify and remove points which reside outside of the target camera's field of view (FOV). The NuScenes dataset provides images and point cloud information for each "snapshot" (i.e., frame) within each 20-s scene. The point cloud data retrieved from the dataset resides in the point sensor frame: in order to determine which points lie within the camera FOV and which points do not; the point cloud must be transformed from the point sensor frame to the image frame. Once the driving data have been collected, well-synchronized keyframes are sampled (image, LIDAR, RADAR) at 2 Hz. These samples are annotated using expert annotators and multiple validation steps in order to produce highly accurate annotations. All objects in the nuScenes dataset comes with a semantic category, as well as a 3D bounding box and attributes for each frame they occur in.

First, the point cloud is rotated and transformed from the point sensor frame to the vehicle ego frame for the timestamp of the relevant LiDAR sweep. This same process is subsequently used to transform the data from the vehicle ego frame to the global frame, from the global frame to the ego vehicle frame, and finally, the ego vehicle frame to the camera plane.

Next, a "snapshot" of the LiDAR point cloud is taken. The resulting 2D matrix is multiplied by the camera intrinsic matrix, and renormalization is applied. Points which lie outside of the camera FOV can now be removed using logical AND functions. A margin of 1 pixel is applied for aesthetic purposes, and we ensure that all points are positioned at least 1 m in front of the sensor in order to exclude points which pertain to the camera casing. This process returns a 2D mask of points, whose IDs are matched with the original 3D point cloud in order to identify and remove the points which lie outside of the camera FOV. Figure 13 illustrates the significance of removing points which fall outside of the camera FOV.

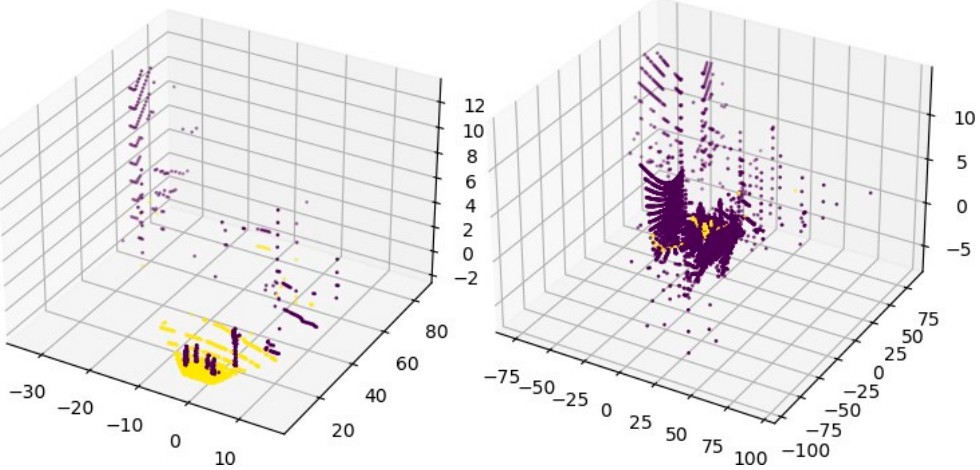

**Figure 13.** RANSAC regression applied to 3D point cloud data.

### 3.6.3. RANSAC Regression

RANSAC (random sample consensus) is a simple, yet highly effective, method of handling outliers through trial and error. It separates data into inliers and outliers, which can be used in further processing techniques. In the context of this paper, a plane on which the majority of points lie is identified with an that this plane is likely to be the ground plane, these points can be removed.

In 3D space, RANSAC functions by first selecting three data points at random from a given point cloud—three points are chosen as this is the minimum number of points required in order to define a plane in 3D space. These identified points are assumed to be inliers and, once the plane has been defined, the number of data points which lie on this plane is tallied. This tally, alongside the points used to define the plane, is stored. This process is repeated for n number of trials or until a plane, on which x% of points lie, has been defined.

Here, a residual threshold of 0.4 is used. The algorithm is halted when a plane is identified which consists of 30% of the total point cloud data, or 50 trials have been conducted. The result is a mask of inlier points, which correspond to those lie on the ground plane; these points are removed from the original point cloud data. Figure 13 illustrates the effectiveness of RANSAC. Yellow points on Figure 13 distinguish the identified ground points and indicate the RANSAC-identified ground plane.

Note that it would be possible for the algorithm to misidentify objects as ground points. Consider the case where a vehicle approaches a T-intersection with a large office building directly ahead; it is likely that the points which make up the office building far outnumber those which lie on the ground and would subsequently be identified as 'ground'. This can be alleviated through the definition of an angle threshold, relative to the vehicle, which must be respected by proposed ground planes (e.g., if a plane is angled at >5° from the vehicle, it is ignored).

### 3.6.4. Clustering Objects

DBSCAN is a widely used clustering technique. It makes use of two parameters: epsilon and minimum points. It works by randomly selecting a point within the dataset from which a potential cluster can be defined. The epsilon is a distance parameter which forms a radius around the selected data point; all other points which fall within this radius are considered "core" points. If the number of core points exceeds the defined number of minimum points, a cluster is initialized. Once a cluster has been initialized, all points which lie within the epsilon of the core points are added to the cluster-these are known as "border" points. Border points are those which are considered to be part of the cluster, but do not lie within the epsilon of the starting point. These border points then project their own radius, which gathers further related points and adds them to the cluster-this process is repeated until no further points are identified. Once a cluster is finalized, and no further points have been identified in an iteration, a new starting point is randomly selected from the remaining points which do not belong to an existing cluster, and the entire process is repeated.

The algorithm developed during this project made use of DBSCAN, with an epsilon of 0.3 and a minimum point's value of 2. The results can be seen in Figure 14a. Individual objects in Figure 14 are distinguished by colour. Each colour indicates a separate DBSCAN-identified cluster. Those on the left are pedestrians, and the yellow cluster to the right is a false detection of a traffic light pole. Figure 14b shows the actual image.

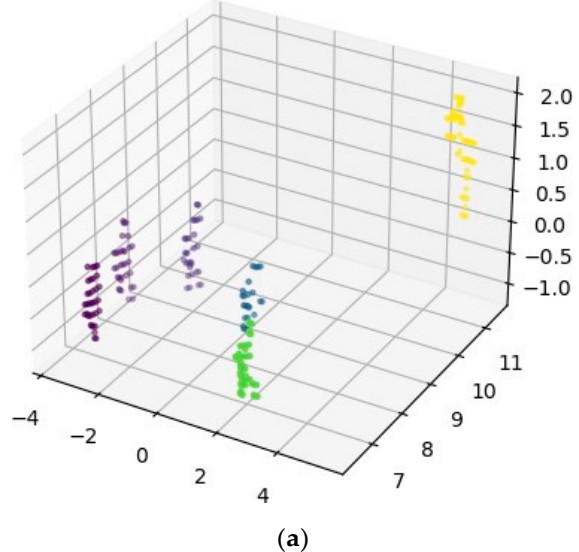

(**a**)

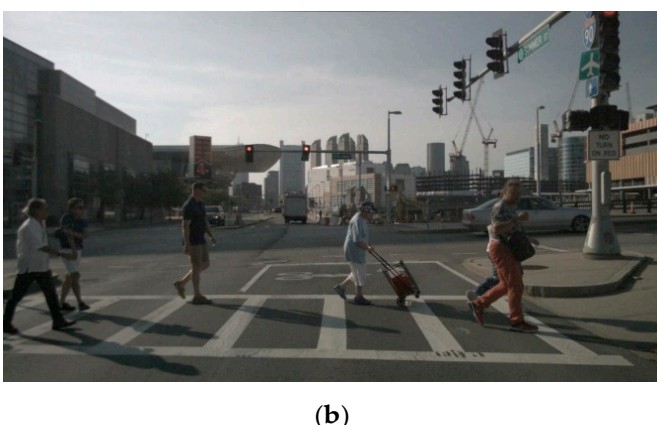

(**b**)

**Figure 14.** (**a**) Clustered objects defined via DBSCAN clustering. (**b**) Actual image.

While the system described here is incapable of classifying pedestrians, it serves as a foundation from which a system, more accurate than the classification models discussed in previous sections, can be developed through sensor fusion.

## 4. Results and Discussion

When summarizing the performance of classification algorithms, a simple metric to use is classification accuracy, which is the total number of correct predictions divided by the total predictions. While this allows for a general overview of how well a model might be performing, it lacks details which can be used to better understand the performance of a model and diagnose where it might be failing. A confusion matrix allows for a better understanding of model performance, directly illustrating the ratio of correct and incorrect predictions for each class. In this section, the use of many confusion matrices—one for each model produced— are utilized to show the effectiveness of the approach.

### 4.1. Rudimentary CNN Models

The two rudimentary models that produced reasonably well results, covered in detail at the start of Section 3, are used as a benchmark to compare the further improved transfer learning models.

The most basic model achieved a classification accuracy of 96% on the test set, with the majority of incorrect classifications belonging to the positive sample class (i.e., pedestrians). Interestingly, an identical model with regularization applied via dropout layers produced almost identical results: It can be seen that the model with regularization performed slightly better in predicting negative samples (i.e., no pedestrian present), however it did misidentify one more positive sample than the most basic model.

It may appear that the addition of dropout resulted in a questionable increase in performance, referring back to the training summary for each model (Figure 4 for the most basic model 6 for the dropout model), there is a significant difference in overfitting-indicated by the disparity between training and validation loss over time. This may be the result of using the same core dataset to compose the training, validation, and testing data splits. It would likely prove beneficial to produce a second testing dataset in order to further investigate the differences between the two models described here.

### 4.2. Transfer Learning Models

Three transfer learning models were developed and discussed below with results presented on the test data.

#### 4.2.1. Transfer Learning Feature Extractor Model

Moving on to Transfer Learning-based models, starting with the VGG-16 feature extraction model. It is seen that this model underperforms compared to the previous two models (Section 4.1). Referring back to the training summary for this feature extractor model, the loss during training is significantly lower than the best performing rudimentary model (Figure 7). This is indicative that the feature extractor model is less overfit than the rudimentary model—the rudimentary model performs better on the test set, however it may be the case that the feature extractor model performs better on data derived from completely new datasets. Furthermore, the fact that convolutional blocks have been frozen during the training of this feature extractor model should also be taken into account; it was unable to adapt to the provided dataset.

#### 4.2.2. Transfer Learning Models with Image Augmentation

As discussed in Section 3.3, it is imperative that suitable image augmentation is applied prior to the training of any classification model. Here, two models which make use of image augmentation at varying intensity are used.

A model trained using overly intense image augmentation significantly underperforms, compared to all other models. Approximately 90% of positive samples (i.e., those

that contain pedestrians) are misidentified as not containing a pedestrian. The model trained with suitable image augmentation, shows a substantial improvement in the model's ability to identify pedestrians in data samples, slightly outperforming the feature extraction model.

### 4.2.3. Transfer Learning Adaptive Model

The adaptive model is the best-performing model described in this paper. It yields an accuracy of approximately 98% on testing data (100% for negative samples, 96.6% on positive samples). Of course, this is due to the 'unfreezing' of convolutional blocks 4 and 5 during training. The ability of this model to adjust its weights during training enabled it to better understand the dataset and adapt accordingly.

Table 2 shows the accuracy of each model developed. All models, with the exception of that which has been trained using flawed image augmentation, perform admirably. The caveat is that, as shown in the training summaries under Section 3.3 all models appear to be overfitting to varying degrees. Further, Table 3 presents the confusion matrix compares the models in terms of additional metrics of precision, recall and F1 score.

**Table 2.** Classification results on the models proposed.

| Model | Percentage Accuracy on Positive Samples | Percentage Accuracy on Negative Samples | Percentage Accuracy on All Samples |
|---|---|---|---|
| Rudimentary CNN model | 92.6 | 99.2 | 96 |
| Rudimentary CNN model with regularization | 99.8 | 92.4 | 96.4 |
| TF model with feature extractor | 100 | 68.2 | 84.1 |
| TF model with flawed image augmentation | 100 | 8.2 | 54.1 |
| TF model with suitable image augmentation | 100 | 66.4 | 83.2 |
| Adaptive TF model | 100 | 96.6 | 98.3 |

**Table 3.** Confusion matrix.

| Model | Precision | | Recall | | F1 Score | |
|---|---|---|---|---|---|---|
| | Positive | Negative | Positive | Negative | Positive | Negative |
| Rudimentary CNN model | 0.99 | 0.93 | 0.93 | 0.99 | 0.96 | 0.96 |
| Rudimentary CNN model with regularization | 1 | 0.93 | 0.92 | 1 | 0.96 | 0.96 |
| TF model with feature extractor | 1 | 0.76 | 0.68 | 1 | 0.81 | 0.86 |
| TF model with flawed image augmentation | 1 | 0.52 | 0.08 | 1 | 0.15 | 0.69 |
| TF model with suitable image augmentation | 1 | 0.75 | 0.66 | 1 | 0.80 | 0.86 |
| Adaptive TF model | 1 | 0.97 | 0.97 | 1 | 0.98 | 0.98 |

The adaptive transfer learning model, the best performing model proposed in this paper, suffers from minimal overfitting. Furthermore, it appears to perform exceptionally well on training data. Rudimentary models, while offering acceptable performance in classification of test data, have not been trained on a dataset as large as those which make use of transfer learning. Comparing the results in [43] where they used three different datasets and trained models using transfer learning, the results in this paper are very similar. The authors achieved an accuracy of 96.71% with 2000 training samples and 99.52% with 5000 training samples using SVM classifiers on the PRID database. In this paper, an accuracy of 98.3% on 3000 training samples is achieved.

As transfer learning models proposed in this paper use the VGG-16 model weights, they can be considered more reliable; while not trained directly on the ImageNet database, the inherent knowledge acquired from the VGG-16 model's training on ImageNet provided a foundation from which a domain-specific (i.e., pedestrian classification) understanding can be cultivated.

### 4.3. Sensor Fusion

In Sections 3.1–3.5, a description how a CNN might be trained for use in a pedestrian classification system is explained, and in Section 3.6, a process for object clustering on LiDAR point cloud data can be applied to segment points which represent pedestrians in the environment surrounding a road vehicle. While CNN methods can provide acceptable classification results for pedestrian detection, the combination of visual and spatial data holds the potential to improve the efficiency and effectiveness of a pedestrian detection system through sensor fusion. There are two categories of sensor fusion: early fusion and late fusion.

Sensor fusion compiles the outputs of multiple sensors, such as LiDAR, RADAR, cameras, etc. The goal of sensor fusion is to create a model which leverages the strengths of each sensor type in the hopes of mitigating their weaknesses. An example of this is the use of LiDAR point cloud data to identify pedestrians in low-light conditions, in which a camera would likely underperform. Conversely, consider a situation where a pedestrian is standing next to a set of traffic lights: a 3D classification system may determine that the pedestrian and traffic lights make up the same object-a camera would be able to differentiate between the two.

In early fusion, the raw data produced by sensors is fused together. LiDAR point cloud data produced by LiDAR sensors can be projected onto 2D images gathered by cameras, for example. 2D object detection on images can be combined with region of interests (ROIs) generated through this point cloud projection through ROI matching, which validates candidate detections proposed by the 2D detection system.

In late fusion, the results of independent detections are fused. LiDAR point clouds are fed into 3D object detection systems, while images are fed into 2D object detection systems. A 3D projection can be created from these 2D detections, which are then cross-referenced with the LiDAR detections via intersection over union (IOU) matching.

### 5. Conclusions

In this paper, a discussion of the existing work pertaining to pedestrian classification through machine learning and deep learning techniques for an autonomous vehicle is presented and a review of convolutional neural networks and how they can be applied in the scope of pedestrian classification is included.

A number of classification models have been proposed, trained on the CVC-02 dataset. It was found that the regularization did not lead to significant improvements in accuracy, however, it did result in a less-overfitting model which is able to better generalize unseen data. Additionally, the image augmentation must be appropriately applied to training data prior to the training of a classification model. Failure to do so will produce a model which significantly underperforms and is unsuitable for use in an autonomous driving system.

The advantage of VGG-16 architecture with a transfer learning model is discussed and shown to have better performance than the models trained using traditional methods. Furthermore, it is concluded that allowing convolutional layers to update their weights during training is beneficial and can lead to exceptional results when compared to models trained with their convolutional layers frozen.

Additionally, a pre-processing system, whereby LiDAR point cloud data is prepared for use in a 3D object classification model, making use of RANSAC regression and DBSCAN clustering, and methods by which visual and spatial data can be combined via sensor fusion in order to boost the results of a pedestrian classification system, are proposed.

**Author Contributions:** Conceptualization, A.K. and A.M.; methodology, A.M.; software, A.M.; validation, A.M.; formal analysis, A.M.; investigation, A.M.; resources, A.M. and A.K.; data curation, A.M.; writing—original draft preparation, A.M.; writing—review and editing, A.M., A.K. and S.S.; visualization, A.M. and A.K.; supervision, A.K.; project administration, A.K.; funding acquisition, A.K. and S.S. All authors have read and agreed to the published version of the manuscript.

**Funding:** This research received no external funding. The submitted research is an outcome of MSc dissertation which was commissioned to produce preliminary results in support of the EU European Regional Development Fund (ERDF) funded project Marine-i. The publication costs will be met by EU ERDF funded Marine-i project.

**Conflicts of Interest:** The authors declare no conflict of interest.

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
