# Peer review of "Deep and Transfer Learning Approaches for Pedestrian Identification and Classification in Autonomous Vehicles"

_electronics, doi:10.3390/electronics10243159_

Round 1

Reviewer 1 Report

In this paper, the authors propose deep learning-based pedestrian identification and classification. The authors provide a detailed introduction to the method and show some results to demonstrate the effectiveness of the identification and classification. There are some questions about this work.

  1. The draft reads like an engineering experiment paper to me by introducing the experiments, existing methods, and the results with the existing methods. I did not observe the algorithm contribution or novelty from the authors. I think this is essential for an academic journal paper. 
  2. In Sec. 2 ‘Review of CNNs for pedestrian recognition, the introduction of ‘single layer perceptrons’ …  ‘linear activation function’... is very basic deep learning concepts. I suggest removing this Section since these are generally understandable by the readers in this journal and make the paper more concise. 
  3. It would be great if the authors can show an example of the fusion between RGB and LiDAR output.
  4. It would be great if the authors can provide visible identification of pedestrians.

Author Response

Dear Reviewer 1,

Thank you for your consideration of our manuscript “Deep and Transfer Learning Approaches for Pedestrian Identification and Classification in Autonomous Vehicles” for publication in the special issue “Autonomous Vehicles Technological Trends” in the Electronics Journal. We have made amendments to the manuscript reflecting the valuable review comments forwarded and feel the paper has been well enhanced to be re-presented for publication. All revisions are added in ‘red’ to make it easier for the reviewers to follow. Please let us know if there is anything we can further do to clarify or extend our paper.

Many thanks,

Kind regards

Asiya

Reviewer 2 Report

1. In Abstract, the authors should state how transfer learning is applied?
2. The current manuscript pipes up a series of deep-learning technologies and models. Please clarify the academic contributions and/or the novelty of the manuscript. It is hard to say the existing contributions (L148-150) are real academic contributions.
3. It is not necessary to introduce all the activation functions in Sec 2.4 – Sec 2.9 one by one.  
4. It is hard to understand the “Predicted Truth” in Fig. 20-23.
5. In Table 2, what does “age accuracy” stand for? I suggest adding more evaluation indicators in the experiments.
6. The experimental results must be compared with some SOTA methods.
7. I suggest adding a detailed network architecture containing all the used technologies to make readers better understand the work in this manuscript.
8. How about the efficiency of the methods.
9. The title contains “Autonomous Vehicles”. How does the proposed work connect to Autonomous Vehicles?
10. The writing must be improved.

Author Response

Dear Reviewer 2,

Thank you for your consideration of our manuscript “Deep and Transfer Learning Approaches for Pedestrian Identification and Classification in Autonomous Vehicles” for publication in the special issue “Autonomous Vehicles Technological Trends” in the Electronics Journal. We have made amendments to the manuscript reflecting the valuable review comments forwarded and feel the paper has been well enhanced to be re-presented for publication. All revisions are added in ‘red’ to make it easier for the reviewers to follow. Please let us know if there is anything we can further do to clarify or extend our paper.

Please see details in the response document attached.

Many thanks

Kind regards

Asiya

Reviewer 3 Report

The authors present learning-based methods for pedestrian classification. The topic is interesting and useful for autonomous driving industry. A few suggestions which may further improve the manuscript. 

  1. The author may consider adding original image for the result shown in Fig. 14, whether the pedestrian is recognized from the method. A heatmap overlaid with original image will be useful to show this (e.g. Grad-CAM)
  2. There are many typos in the manuscript. Please proofread carefully. 
  3. The authors introduced a lot of fundamental information of deep learning, which may not necessary.
  4. The authors may consider adding samples regarding how they preprocess the images in section 3.6.2.
  5. It may not necessary to use plots for Fig. 20, 21 and 22. Using tables may be enough to show the same information. 

Author Response

Dear Reviewer 3,

Thank you for your consideration of our manuscript “Deep and Transfer Learning Approaches for Pedestrian Identification and Classification in Autonomous Vehicles” for publication in the special issue “Autonomous Vehicles Technological Trends” in the Electronics Journal. We have made amendments to the manuscript reflecting the valuable review comments forwarded and feel the paper has been well enhanced to be re-presented for publication. All revisions are added in ‘red’ to make it easier for the reviewers to follow. Please let us know if there is anything we can further do to clarify or extend our paper.

Please see the uploaded word document.

Many thanks

Kind regards

Asiya

Round 2

Reviewer 2 Report

The revisions have improved the manuscript. I think that it can be accepted for publication now. 

Author Response

We thank the reviewer for their time and consideration in the review of our manuscript. It has considerably improved our work.